# Regulation of the Macroautophagic Machinery, Cellular Differentiation, and Immune Responses by Human Oncogenic γ-Herpesviruses

**DOI:** 10.3390/v13050859

**Published:** 2021-05-08

**Authors:** Christian Münz

**Affiliations:** Laboratory of Viral Immunobiology, Institute of Experimental Immunology, University of Zürich, 8057 Zürich, Switzerland; christian.muenz@uzh.ch

**Keywords:** Epstein-Barr virus (EBV), Kaposi sarcoma-associated herpesvirus (KSHV), lymphomas, immune escape, T cell responses, co-infections, humanized mice

## Abstract

The human γ-herpesviruses Epstein-Barr virus (EBV) and Kaposi sarcoma-associated herpesvirus (KSHV) encode oncogenes for B cell transformation but are carried by most infected individuals without symptoms. For this purpose, they manipulate the anti-apoptotic pathway macroautophagy, cellular proliferation and apoptosis, as well as immune recognition. The mechanisms and functional relevance of these manipulations are discussed in this review. They allow both viruses to strike the balance between efficient persistence and dissemination in their human hosts without ever being cleared after infection and avoiding pathologies in most of their carriers.

## 1. Introduction on Human Oncogenic γ-Herpesviruses

Herpesviruses contain some of the most successful pathogens in humans, some of which persistently infect more than half of the adult population [1]. The human γ-herpesviruses, namely Epstein-Barr virus (EBV) or human herpesvirus 4 (HHV4) and Kaposi sarcoma-associated herpesvirus (KSHV) or human herpesvirus 8 (HHV8), are no exception [2,3,4]. They persist in more than 90% of the human adult population in the case of EBV and in more than 50% in the case of KSHV in sub-Saharan Africa [4,5]. Contrary to α- and β-herpesviruses, EBV and KSHV are lymphotropic and are thought to primarily colonize long-lived differentiation stages of human B cells [6]. Both are transmitted via saliva exchange and their entry receptor into epithelial cells, the ephrin receptor A2 [7,8,9], is expressed in the oropharyngeal mucosa. EBV and KSHV might gain access to submucosal secondary lymphoid tissues without epithelial cell infection by transcytosis or similar mechanisms [10,11]. EBV is then thought to stimulate B cell differentiation after infection via latent gene expression [12]. In naïve B cells of healthy virus carriers, six nuclear antigens of EBV (EBNAs), two latent membrane proteins (LMPs), two EBV-encoded small non-translated RNAs (EBERs) and more than 40 viral miRNAs of the BHRF1 and BART clusters are expressed [3]. This latency III infection program is also induced upon primary human B cell infection with EBV in vitro, resulting in immortalized lymphoblastoid cell lines (LCLs). In vivo differentiation of activated naïve B cells into germinal center B cells is associated with the downregulation of latent EBV gene expression to EBNA1, the two LMPs, EBERs and BART miRNAs [13,14]. This latency II infection program is thought to rescue germinal center B cells in order for the virus-infected cells to gain access to the memory B cell pool for long-term maintenance [15]. In these memory B cells, EBV switches off all viral protein expression and only non-translated EBERs and BART miRNAs can be found. This program is called latency 0. Upon their homeostatic proliferation, the viral episomal DNA maintenance protein EBNA1 is transiently expressed in memory B cells, constituting latency I [16]. EBV latency is the default program upon B cell infection because lytic replication to produce infectious virus particles can only occur from circularized episomal latent viral DNA after its epigenetic modification with extensive methylation [17]. This takes around two weeks after EBV infection of B cells and can primarily be induced from latency 0 and I by, for example, B cell receptor stimulation and associated plasma cell differentiation [18]. Physiological infection of B cells by KSHV is much less well understood, but recent evidence suggests that this EBV-related γ-herpesvirus might profit from EBV infection. Along these lines, EBV co-infection has been shown to support KSHV persistence in both mice with reconstituted human immune system compartments (humanized mice) and in vitro [19,20,21,22]. Accordingly, EBV infection seems to accompany KSHV infection in different African child cohorts [23,24]. It is tempting to speculate that the earlier saturation of African populations with persistent EBV infection might support the increased seroprevalence of KSHV infection in sub-Saharan Africa [4,5]. Under which conditions KSHV reactivates and infects endothelial cells from which Kaposi sarcoma originates is, however, unclear. Nevertheless, lytic reactivation of both EBV and KSHV in submucosal tissues might lead to oropharyngeal epithelial infection followed by lytic replication to amplify shedding into saliva for further transmission [25,26]. For EBV, this basolateral infection of polarized epithelia seems to be at least in part mediated by BMRF2 binding to αv and β1 integrins [10] and/or direct contact with EBV-infected B cells [27].

In addition to their successful colonization of humans, EBV and KSHV are also potent oncoviruses and designated as class I carcinogens by the WHO [28,29,30]. Curiously, the very same infection programs as outlined above are also found in EBV- and KSHV-associated malignancies, with EBV latency III in a subset of diffuse large B cell lymphomas (DLBCLs) or post-transplant lymphoproliferative disease (PTLDs), EBV latency II in Hodgkin’s lymphoma and nasopharyngeal carcinoma, EBV latency I in Burkitt’s lymphoma and EBV plus KSHV co-infection in primary effusion lymphoma (PEL) [31]. All EBV- and KSHV-associated B cell lymphomas and Kaposi sarcoma increase in frequency during immune suppression, such as during immune modulatory treatment after transplantation or after co-infection with the human immunodeficiency virus (HIV). Thus, premalignant persistent EBV and KSHV infections are harbored by large proportions of the human population and seem to be immune controlled for life in most individuals. How these two human oncogenic γ-herpesviruses manipulate cellular processes to strike this balance between oncogenesis, persistence and immune control will be discussed in this review.

## 2. Regulation of Macroautophagy Proteins by EBV and KSHV

Despite their collaboration for KSHV persistence and during PEL lymphomagenesis, EBV and KSHV influence some cellular pathways in opposite directions. One of these pathways is the pro-survival catabolic function of macroautophagy. Macroautophagy is one of at least three mechanisms by which cytoplasmic material can be delivered to lysosomes for degradation [32]. At least 40 autophagy-related (Atg) proteins contribute to the generation of double membrane-surrounded vesicles and the fusion of these autophagosomes with lysosomes [33]. Four stages in this process can be distinguished (Figure 1). The first one involves the activation of the Ulk1/Atg1 protein kinase complex that is inhibited by the mechanistic target of rapamycin (mTOR) and activated by AMP-activated protein kinase (AMPK) during nutrient deficiency. The Ulk1/Atg1 complex regulates many steps during macroautophagy [34], including activation of the VPS34 and Beclin-1/Atg6-containing phosphoinositide 3-kinase (PI3K) complex that constitutes the second stage of macroautophagy. The resulting PI3P on membranes, often in the endoplasmic reticulum, recruits the third stage, namely molecular machinery that covalently links ubiquitin-like molecules to membranes that will become autophagosomes. For this purpose, WIPI2 binds both PI3P and the Atg5/Atg12/Atg16L1 complex that serves as an E3-like ligase for the ubiquitin-like proteins LC3A, LC3B, LC3C, GABARAP, GABARAP-L1 and GABARAP-L2 to phosphatidylethanolamine. For this conjugation reaction, these ubiquitin-like molecules are first C-terminally trimmed by Atg4 proteases, then activated by the E1-like Atg7 and transferred by the E2-like conjugating enzyme Atg3 to the Atg5/Atg12/Atg16L1 complex for lipidation. The Atg5/Atg12/Atg16L1 complex is also assembled by a ubiquitin-like reaction during which Atg12 gets activated by Atg7 and then coupled to Atg5 via the E2-like conjugating enzyme Atg10. The conjugation of LC3s and GABARAPs to membranes allows for elongation of the isolation membrane that will become the autophagosome and recruitment of substrates into this vesicle. Substrate recruitment can be mediated by autophagy receptors that contain LC3-interacting regions (LIRs) and bind substrates, including ubiquitinated protein aggregates. Sequestosome 1/p62, NBR1, NDP52 and TAX1BP1 are some of these receptors [35]. Once autophagosomes are completed by this machinery, Atg4 proteases cleave LC3s and GABARAPs from the outer autophagosomal membrane and fusion with lysosomes is the fourth stage. Rab7, syntaxin 17 and YKT6 are involved in this fusion process [36,37] that leads to degradation of the autophagosome cargo and the inner autophagosomal membrane that is still decorated with LC3s and GABARAPs. Nutrient recycling from autolysosomes allows for cellular survival. 

In addition, the molecular machinery of macroautophagy is highly attractive for viruses because it facilitates membrane remodeling and can also degrade viral capsids in the cytosol [38]. While EBV utilizes this machinery during latent and lytic infection, KSHV primarily focusses on inhibiting its degradation via macroautophagy (Figure 1). During latent EBV infection, LMP1 stimulates macroautophagy to modulate its degradation and thereby adjust its expression levels [39]. In addition, EBNA3C seems to stimulate macroautophagy for B cell survival [40]. Indeed, macroautophagy seems to sustain latent EBV infection, while lytic reactivation increases upon macroautophagy inhibition [41]. In contrast to regulation of autophagosome formation during latent infection, during lytic EBV infection, autophagic membranes are stabilized and inhibited in their fusion with lysosomes [42,43]. This favors infectious viral particle release and LC3B-conjugated membranes have been found in the EBV envelope [42,43]. At the same time, the virus blocks, with the lytic gene products BHRF1 and BPLF1, mitophagy and selective macroautophagy via p62 [44,45], possibly in order to preserve LC3B-coupled membranes for EBV envelope acquisition. These findings suggest that EBV uses the macroautophagy machinery for its own benefit, stabilizing latent infection by macroautophagy induction and supporting infectious virus production by autophagosomal membrane stabilization as well as integration into its envelope.

In contrast, KSHV seems to inhibit macroautophagy to escape its anti-viral functions, but requires it for lytic reactivation [46]. Both latent and lytic KSHV gene products have been implicated in this macroautophagy inhibition, affecting both autophagosome formation and fusion with lysosomes. Along these lines, the viral Bcl-2 (vBcl-2) ortholog encoded by orf16 blocks macroautophagy by Beclin-1 binding [47]. In contrast, Beclin-2 directs the viral G-protein-coupled receptor (vGPCR) encoded by orf74 to endolysosomal degradation and thereby limits vGPCR-mediated Kaposi sarcoma-like pathogenesis in mice [48]. Furthermore, the viral FLIP (vFLIP) encoded by orf71 inhibits Atg3 [49,50]. After autophagosome formation, KSHV also inhibits fusion with lysosomes via binding of its K7 protein to Rubicon, a negative regulator of autophagosome maturation [51]. Furthermore, KSHV infection downregulates, with Rab7, a component of the fusion machinery with lysosomes [46,52]. Thus, EBV seems to utilize the macroautophagy machinery, while KSHV primarily inhibits its anti-viral function. Future studies will need to characterize if these different viral mechanisms to modulate the macroautophagy machinery synergize or antagonize each other during co-infection.

## 3. Influence of Human γ-Herpesviruses on B Cell Differentiation and Lymphomagenesis

The two human γ-herpesviruses encode oncogenes that, when transgenically expressed in mice, cause B cell lymphomas either on their own [53,54,55] or together with cellular oncogenes [56]. Accordingly, EBV infection of human B cells proceeds in three stages to transformed LCLs [57,58,59,60,61,62]. In the first stage, leaky expression of lytic EBV gene products, including the two anti-apoptotic viral Bcl-2 proteins BALF1 and BHRF1, is required to ensure survival of infected cells [57]. Then, a rapid proliferation phase starts and is driven by EBNA2- and EBNA-LP-induced expression of the c-myc oncogene [63]. This proliferative phase is vulnerable to apoptosis induction by the DNA damage response [64], but is rescued by EBNA3A- and EBNA3C-mediated suppression of p16 (INK4a) and Bim [65,66,67]. This infection program has also been named latency IIb [60] and precedes expression of LMP1 and LMP2, which take over proliferation stimulation and protection from apoptosis only after several days of EBV infection for complete LCL establishment and latency III [68,69]. LMP1 and LMP2 then mimic signaling of CD40 co-stimulation and B cell receptor signaling, respectively [70]. The germinal center reaction seems to allow for the transition from latency III to latency II and finally latency 0 [13], possibly under the influence of IL-21 that is produced by follicular helper T cells [71,72]. Interestingly, long-term persistence of EBV with latency 0 can be achieved without going through latency III, but possibly directly from latency IIb if EBNA3C is missing [73]. Thus, EBV has different stages of B cell activation after infection that are thought to provide access for EBV-positive naïve B cells to the germinal center reaction and rescue them into the memory B cell pool. This differentiation has been discussed more extensively in recent reviews [2,3]. In this process, LMP1 is considered to be the main viral oncogene and its transgenic expression in mouse B cells causes lymphomas [53].

In contrast, KSHV does not transform B cells in vitro and transgenic expression of its latent gene products LANA, vFLIP, vCyclin, kaposin and viral miRNAs in mice also does not lead to the development of B cell lymphomas [74]. However, in the presence of cellular oncogene expression, such as c-myc, these latent KSHV gene products can augment lymphomagenesis [56]. Furthermore, they drive B cells into plasma cell differentiation [74] which is also a hallmark of PELs [75]. Interestingly KSHV co-infection also increases this plasma cell differentiation in EBV-immortalized B cells [19]. This plasma cell differentiation then stimulates BZLF1 expression and reactivation of lytic EBV replication via the transcription factors XBP1 and BLIMP1 [76,77]. Lytic EBV infection then further drives plasma cell differentiation [78]. Consistent with these findings, lytic EBV replication can also be found in plasma cells of healthy EBV carriers [18]. Surprisingly, however, this elevated lytic EBV infection due to KSHV co-infection promotes lymphomagenesis [19]. Additionally, not only during KSHV co-infection, but also in other EBV-associated tumors, lytic EBV replication in a subset of cells seems to condition the microenvironment for malignancies [3,79,80,81,82,83,84,85]. Thus, KSHV co-infection drives plasma cell differentiation of EBV-infected B cells, augmenting lytic EBV replication in a subset and thereby lymphomagenesis.

## 4. Manipulation of Antigen Presentation and Immune Recognition by EBV and KSHV

In addition to these modifications to cellular processes that ensure persistence of human γ-herpesvirus infections which mainly influence cellular proliferation and protection from cell death, EBV and KSHV have evolved strategies to escape immune eradication (Figure 2). These strategies involve both immune evasion proteins and miRNAs that compromise immune activation as well as antigen presentation [86]. In this respect, early lytic EBV proteins block MHC class I-restricted antigen presentation to CD8^+^ T cells [87,88]. For this purpose, BNLF2a blocks the transporter associated with antigen presentation (TAP) that imports cytosolic peptides, often produced by proteasomal degradation, into the endoplasmic reticulum for loading onto MHC class I molecules [89,90,91,92]. In addition, the G-protein-coupled receptor BILF1 internalizes MHC class I molecules for degradation [93]. Finally, the host shutoff protein BGLF5 compromises MHC class I-restricted antigen presentation [94,95]. In comparison, BNLF2a primarily compromises immediate early MHC class I presentation, for example, BZLF1, and early MHC class I presentation, for example, BMRF1, EBV antigens, while BILF1 rather inhibits recognition of early and late EBV antigens [96]. Apart from these immune evasions and also affecting, in addition to lytic replication, latencies IIb and III, presumably primarily BHRF1 miRNAs downregulate MHC class I-restricted antigen presentation by decreasing TAP2 [97]. This indeed results in improved immune control of miRNA-deficient EBV in humanized mice which is primarily mediated by CD8^+^ T cells [98]. Finally, EBNA1 blocks its own proteasomal degradation and ribosomal translation via its glycine–alanine repeat domain and thereby escapes MHC class I presentation [99,100,101]. Similar to EBV, KSHV encodes K3 and K5 that direct MHC class I molecules for degradation [102,103,104]. These genes encode ubiquitin ligases that stimulate MHC class I internalization via ubiquitination [105,106]. Thus, both EBV and KSHV compromise cytotoxic CD8^+^ T cell responses that are also identified by primary immunodeficiencies as the main protective immune entities at least during EBV infection [107] and whose depletion leads to loss of immune control in EBV-infected humanized mice [98,108,109,110].

Immune evasion by EBV and KSHV, however, also affects MHC class II-restricted antigen presentation to helper CD4^+^ T cells (Figure 2). The late lytic EBV protein gp42 also binds to MHC class II during viral entry [111] but then also interferes with MHC class II recognition by the T cell receptor [112,113]. Furthermore, viral miRNAs downregulate components of lysosomal proteolysis, such as the cysteine protease legumain (LGMN) and the γ-interferon-inducible lysosomal thiol reductase (GILT), and thereby compromise MHC class II ligand generation for CD4^+^ T cell recognition [114]. KSHV can also downregulate MHC class II molecules by K3- and K5-dependent ubiquitination [115]. Moreover, viral interferon regulatory factor 3 (vIRF3) of KSHV attenuates MHC class II expression [116]. Thus, both human γ-herpesviruses invest some effort to compromise CD4^+^ T cell recognition, and CD4^+^ T cell depletion by antibodies or HIV co-infection decreases EBV-specific immune control in humanized mice [108,110]. However, the importance of CD4^+^ T cells for EBV-specific immune control is not that straightforward because patients with MHC class II deficiencies usually do not present with EBV-associated pathologies [117]. 

In contrast, another cytotoxic lymphocyte population that seems to play a role in the immune control of lytic EBV replication is natural killer (NK) cells [118,119,120,121]. Their deficiency also predisposes for EBV-associated pathologies [122,123,124]. An important NK cell receptor in the recognition of EBV lytically replicating cells is NKG2D [125]. Interestingly, both EBV and KSHV downregulate NKG2D ligands (Figure 2) that in humans belong to the MIC (A and B) or ULBP (1–6) molecules. Surprisingly, both EBV and KSHV focus on the downregulation of MIC molecules [126,127]. Both EBV- and KSHV-encoded miRNAs compromise MICB expression [126], while K5 of KSHV ubiquitinates both MICA and MICB for internalization and degradation [127]. Thus, both viruses seem to compromise NK cell recognition, even so their co-infection leads to the differentiation of CD56-negative NK cells with poor function [128].

In addition to these cell intrinsic immune evasion functions, KSHV and EBV modulate their microenvironment by secreted viral cytokines (vIL-10 and vIL-6) and downregulation of both proinflammatory cytokine and chemokine secretion. EBV’s viral IL-10 is expressed from the lytic BCRF1 gene [92,129,130,131,132]. It inhibits MHC class I-restricted antigen presentation and T cell responses. In contrast, viral IL-6 and two viral macrophage inflammatory protein 1-like proteins (vMIPs) of KSHV might recruit and stimulate immune-suppressive myeloid cell populations to the vicinity of KSHV-infected cells [133,134] and might prevent NK cell migration [135]. In addition, EBV miRNAs downregulate the expression of IL-12 and CXCL11 [114,136]. Thus, immune evasion mechanisms of EBV and KSHV modify both cellular pathways that lead to immune recognition of infected cells as well as render their microenvironment less immune stimulatory.

## 5. Conclusions and Outlook

γ-Herpesviruses are unique among the herpesviruses in that they have a separate infection program that allows them to replicate via cellular proliferation and not just via infectious particle production. However, this advantage comes at the price of being detectable to the immune system due to the expression of viral oncoproteins to drive infected B cell expansion and via DNA damage responses that are increased in the proliferating cells. In part, KSHV and EBV also support this immune detection and EBV even encodes with EBNA3B a latent viral protein that induces chemokines (CXCL9 and CXCL10) to attract immune cells via their chemokine receptor CXCR3 into the environment of infected cells in order to preserve its host for asymptomatic persistence with transmission to new hosts [137]. For this purpose, EBV and KSHV have to, however, strike a fine balance of preventing pathology by their oncogenes while escaping clearance by the immune system. Accordingly, they manipulate cellular pathways like macroautophagy, cellular proliferation and apoptosis for replication and at the same time tune immune control to just the right level to allow persistence without damage to the host.

This balance that has developed during the co-evolution of γ-herpesviruses with their human host offers insights into the requirements for both B cell activation and survival in the main host cells of EBV and KSHV, as well as the sustained cell-mediated immune control of these infected cells. These insights provide targets for lymphoma therapy and characteristics of long-lasting protective T cell immunity that would also be desirable in other tumor settings. While these features are better understood for the treatment of latency III DLBCL and can be explored for PELs in EBV with KSHV double infection models, models for latency I Burkitt’s and latency II Hodgkin’s lymphoma, as well as nasopharyngeal carcinoma and Kaposi sarcoma, are still lacking. While the existing models can already be used to develop new therapeutic approaches against EBV- and KSHV-associated malignancies, they could then also be applied to newly developed models for Burkitt’s lymphoma, Hodgkin’s lymphoma, nasopharyngeal carcinoma and Kaposi sarcoma. 

## Figures and Tables

**Figure 1 viruses-13-00859-f001:**
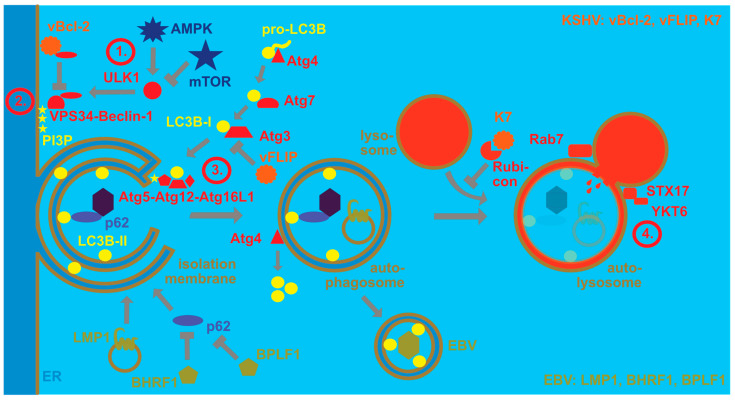
EBV and KSHV regulate the macroautophagy machinery for their own benefit. Autophagosomes form upon nutrient depletion, whereby AMP-activated protein kinase (AMPK) activates, and mechanistic target of rapamycin (mTOR) inhibits the protein kinase ULK1. ULK1 then phosphorylates the PI3 kinase complex containing VPS34 and Beclin-1 (stage 1). This complex phosphorylates PI3 (stage 2). The resulting PI3P recruits the E3-like LC3B ligase complex of Atg5, Atg12 and Atg16L1, which couples LC3B-I to phosphatidylethanolamine (LC3B-II) in the isolation membrane that often originates from the endoplasmic reticulum (stage 3). LC3B-I is formed by pro-LC3B being cleaved by Atg4, then activated by the E1-like enzyme Atg7 and conjugated by the E2-like enzyme Atg3. LC3B-II is involved in the elongation of the isolation membrane until completion of the double-membrane-surrounded autophagosome, as well as substrate recruitment, exemplified by viral capsid recruitment with the macroautophagy receptor p62. Upon completion, LC3B is recycled from the outer autophagosomal membrane by the protease Atg4. The completed autophagosome then fuses with late endosomes or lysosomes for degradation of its content and the inner autophagosomal membrane (stage 4). Rab7, syntaxin 17 (STX17) and YKT6 participate in this fusion to generate the autolysosome. Rubicon inhibits this autophagosome maturation. EBV stimulates macroautophagy via LMP1 which then regulates its cellular levels by degradation via macroautophagy. Furthermore, BHRF1 and BPLF1 inhibit selective macroautophagy via p62. EBV then utilizes LC3B-coupled membranes for its envelope for efficient egress and LC3B-II can be found in purified EBV particles. KSHV blocks macroautophagy by inhibiting Beclin-1 with its viral Bcl-2 homolog (vBcl-2). Furthermore, it blocks Atg3 with its viral FLIP protein (vFLIP). Finally, K7 inhibits autophagosome maturation by binding to Rubicon.

**Figure 2 viruses-13-00859-f002:**
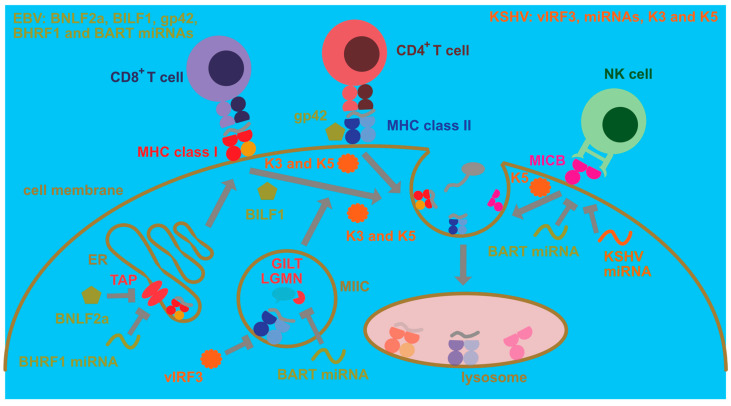
Cell intrinsic immune escape mechanisms by EBV and KSHV. EBV compromises antigen presentation to CD8^+^ T cells via MHC class I molecules by blocking their peptide loading through inhibition of the transporter associated with antigen presentation (TAP) in the endoplasmic reticulum (ER). This TAP inhibition is performed by the BNLF2a protein and by downregulation of TAP2 by BHRF1 miRNAs. Furthermore, BILF1 causes MHC class I complexes with peptides to be internalized from the cell membrane for lysosomal degradation. Moreover, EBV blocks antigen presentation to CD4^+^ T cells by its MHC class II-binding protein gp42. Furthermore, its BART miRNAs block expression of components of lysosomal antigen processing for MHC class II presentation, such as γ-interferon-inducible lysosomal thiol reductase (GILT) and the cysteine protease legumain (LGMN). The BART cluster also contains a miRNA that downregulates MICB, one of the ligands for the activating NK cell receptor NKG2D which is important for both NK and CD8^+^ T cell recognition of EBV-infected B cells. KSHV encodes the two ubiquitin ligases K3 and K5 which cause internalization for lysosomal degradation of MHC class I and II peptide complexes from the cell membrane. K5 can also internalize the NKG2D ligands MICA and MICB. Furthermore, vIRF3 of KSHV inhibits MHC class II expression, and KSHV miRNAs downregulate MICB.

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
