# Peer review of "Regulation of the Macroautophagic Machinery, Cellular Differentiation, and Immune Responses by Human Oncogenic γ-Herpesviruses"

_viruses, 2021, doi:10.3390/v13050859_

Round 1
Reviewer 1 Report
The paper by Munz describes in detail mechanisms adopted by gamma-herpesviruses to manipulate cellular pathways.
I only have minor comments to this review:
L24: please add a ref. to support the data on the percentages cited.
L28: please replace the comma with a period ( between mucosa and EBV)
L: 39-40: you briefly explain the different types of latency, but latency I. Please add a brief sentence also for the latency I infection.
L42:add a comma between 'proliferation' and 'the viral'
L50: please use a more scientific term instead of piggy-bag
L64: the term 'oncoviruses' is more appropriate than 'tumorviruses'
L69-72: check grammar
L91-93: in my opinion, the sentence is unclear
L167: replace 'that' with 'and'
L220: typing error (domain)
L222: Please add 'genes' (These genes encode...)
Figures: both figures are understandable, but in my opinion the choice of color makes the visualization difficult. Maybe changing colors or adding background or edges to the text and arrows will help.
From my point of view, the manuscript by Munz, after these minor revision, deserves publication in Viruses.
Author Response
I thank the reviewers for their constructive comments which I have now incorporated into the revised manuscript. I outline the respective changes in the response to the individual reviews below as well as by underlining in the revised manuscript.
Reviewer #1
The paper by Munz describes in detail mechanisms adopted by gamma-herpesviruses to manipulate cellular pathways.
I only have minor comments to this review:
L24: please add a ref. to support the data on the percentages cited.
The respective references for the seroprevalences of EBV and KSHV have been added.
L28: please replace the comma with a period (between mucosa and EBV)
This has been corrected.
L: 39-40: you briefly explain the different types of latency, but latency I. Please add a brief sentence also for the latency I infection.
This has now been added to L44.
L42: add a comma between 'proliferation' and 'the viral'
A comma has been added to this sentence.
L50: please use a more scientific term instead of piggy-bag
This has now been replaced with ‘profit from’ in L51.
L64: the term 'oncoviruses' is more appropriate than 'tumorviruses'
‘Tumorviruses’ has been replaced with ‘oncoviruses’.
L69-72: check grammar
This sentence has been considerably shortened and simplified.
L91-93: in my opinion, the sentence is unclear
This sentence has been simplified.
L167: replace ‘that’ with ‘and’
This has been changed.
L220: typing error (domain)
This typo has been corrected.
L222: Please add 'genes' (These genes encode...)
‘Genes’ has been added.
Figures: both figures are understandable, but in my opinion the choice of color makes the visualization difficult. Maybe changing colors or adding background or edges to the text and arrows will help.
Some adjustments have been made to the figures.
From my point of view, the manuscript by Munz, after these minor revision, deserves publication in Viruses.
Reviewer 2 Report
This review about a very complex subject is well written. The authors give, in great detail ,the regulatory programmes of the two viruses, EBV and KSHV and how they control the immune mechanism of the host and thus remain persistent. It is interesting to note that KSHV which has been accepted as being involved with HIV, can also co-exist and augment EBV replication. This review mainly focuses on the latent cycles of the two viruses. A review on viral products of these two viruses in oncogenesis may be of interest to the author. Charostad et al . Infectious Agents and Cancer (2020) 15:62.
Author Response
I thank the reviewers for their constructive comments which I have now incorporated into the revised manuscript. I outline the respective changes in the response to the individual reviews below as well as by underlining in the revised manuscript.
Reviewer #2
This review about a very complex subject is well written. The authors give, in great detail ,the regulatory programmes of the two viruses, EBV and KSHV and how they control the immune mechanism of the host and thus remain persistent. It is interesting to note that KSHV which has been accepted as being involved with HIV, can also co-exist and augment EBV replication. This review mainly focuses on the latent cycles of the two viruses. A review on viral products of these two viruses in oncogenesis may be of interest to the author. Charostad et al., Infectious Agents and Cancer (2020) 15:62.
I thank this reviewer for his/her encouraging comments and have now cited the respective review in line 65 on page 2.
Reviewer 3 Report
The author of this manuscript report how human oncogenic γ-herpesviruses Epstein-Barr virus and Kaposi sarcoma associated herpesvirus regulate and manipulate macroautophagy proteins, influence B-cell differentiation and lymphogenesis and evade immune recognition. These viruses must balance these mechanisms in order for the virus to proliferate within the host undetected but also not harm the host. Models of these mechanisms in different levels of infection/latency are currently being used to develop treatment approaches and may also be of use to help develop new models and treatment approaches for premalignant persistent EBV and KSHV infections.
The is a useful and comprehensive review. However, it needs substantial editing / rewriting to make it readable and understandable before it can be accepted.
Some concerns that need to be addressed are:
- In the section on macrophagy, please include a discussion of the interaction between Beclin 2 & KSHV GPCR, and consequent regulation of tumorigenesis (PNAS Mar 2016, 113 (11) 2994-2999; DOI: 10.1073/pnas.1601860113).
- Please remove the blue background of the figures. It makes it difficult to see the text and symbols and serves no useful purpose.
- The manuscript requires significant editing and rewriting. There are many long, run-on sentences, and other sentences that require insert of commas to help with the reading flow of the paper. Some words are overused (such as “for example”) and there are numerous grammatical and language errors.
- In several cases it is unclear whether the protein being referred to is a viral protein or human host protein. Please make sure that is clear. Not all readers will be experts on herpesvirues.
- There are too many abbreviations. If that cannot be avoided, please make sure to define them at first use. Not all of them are defined.
- In the macroautophagy section, “PI3” should be “PI3P”.
- Since macroautophagy is described as having 4 stages, figure 1 could be altered to better display and distinguish between the 4 stages.
- Figure 1 shows the proteins that play a role in macroautophagy except for Orf16 and Orf71 which are the genes which encode v-Blc2 and v-FLIP proteins, respectively. It is the protein interactions which inhibit the autophagy mechanism not the genes.
- A figure showing the mechanism of B-cell differentiation and lymphomagenesis would be helpful in section 3.
- GILT and LGMN shown and introduced in Figure 2 are never mentioned in the actual review.
- When MIC molecules are introduced in line 259 it is stated MIC (A and 6). Later they focus on MICB, is this in the same classification of MIC molecules missing from the previous sentence or are MICB and MIC6 the same?
Author Response
I thank the reviewers for their constructive comments which I have now incorporated into the revised manuscript. I outline the respective changes in the response to the individual reviews below as well as by underlining in the revised manuscript.
Reviewer #3
The author of this manuscript report how human oncogenic γ-herpesviruses Epstein-Barr virus and Kaposi sarcoma associated herpesvirus regulate and manipulate macroautophagy proteins, influence B-cell differentiation and lymphogenesis and evade immune recognition. These viruses must balance these mechanisms in order for the virus to proliferate within the host undetected but also not harm the host. Models of these mechanisms in different levels of infection/latency are currently being used to develop treatment approaches and may also be of use to help develop new models and treatment approaches for premalignant persistent EBV and KSHV infections.
This is a useful and comprehensive review. However, it needs substantial editing / rewriting to make it readable and understandable before it can be accepted.
Some concerns that need to be addressed are:
- In the section on macroautophagy, please include a discussion of the interaction between Beclin 2 & KSHV GPCR, and consequent regulation of tumorigenesis (PNAS Mar 2016, 113 (11) 2994-2999; DOI: 10.1073/pnas.1601860113).
This publication has now been discussed and cited on page 4 of the revised manuscript.
- Please remove the blue background of the figures. It makes it difficult to see the text and symbols and serves no useful purpose.
In would like to keep the blue background because the yellow labels for membrane modifications during macroautophagy are otherwise difficult to read.
- The manuscript requires significant editing and rewriting. There are many long, run-on sentences, and other sentences that require insert of commas to help with the reading flow of the paper. Some words are overused (such as “for example”) and there are numerous grammatical and language errors.
Long sentences have been simplified and the use of for example restricted.
- In several cases it is unclear whether the protein being referred to is a viral protein or human host protein. Please make sure that is clear. Not all readers will be experts on herpesviruses.
This has been clarified in many instances.
- There are too many abbreviations. If that cannot be avoided, please make sure to define them at first use. Not all of them are defined.
Abbreviations whose expansions provide useful information have been explained at first use.
- In the macroautophagy section, “PI3” should be “PI3P”.
This has been corrected when referring to the product of the VPS34 kinase and retained as PI3 when it indicates the substrate of the VPS34 kinase.
- Since macroautophagy is described as having 4 stages, figure 1 could be altered to better display and distinguish between the 4 stages.
Numbers for the four stages have now been inserted in figure 1.
- Figure 1 shows the proteins that play a role in macroautophagy except for Orf16 and Orf71 which are the genes which encode v-Blc2 and v-FLIP proteins, respectively. It is the protein interactions which inhibit the autophagy mechanism not the genes.
This has now been done for vBcl-2 and vFLIP.
- A figure showing the mechanism of B-cell differentiation and lymphomagenesis would be helpful in section 3.
I have now referred to recent reviews which discuss in more detail B cell differentiation in relation to EBV infection and associated lymphomagenesis on page 5 of the revised manuscript.
- GILT and LGMN shown and introduced in Figure 2 are never mentioned in the actual review.
These are now mentioned in the review text on page 6 of the revised manuscript as viral miRNA regulated lysosomal components.
- When MIC molecules are introduced in line 259 it is stated MIC (A and 6). Later they focus on MICB, is this in the same classification of MIC molecules missing from the previous sentence or are MICB and MIC6 the same?
This typo has been corrected to MIC (A and B).